# Motion Characteristics of High-Speed Supercavitating Projectiles Including Structural Deformation

Chuang Huang [1],*, Zhao Liu [1], Zixian Liu [2], Changle Hao [1], Daijin Li [1] and Kai Luo [1]

1   School of Marine Science and Technology, Northwestern Polytechnical University, Xi'an 710065, China; 2020260814@mail.nwpu.edu.cn (Z.L.); jhhaochangle@mail.nwpu.edu.cn (C.H.); lidaijin@nwpu.edu.cn (D.L.); luokai72@163.com (K.L.)
2   AVIC Xi'an Automatic Control Research Institute, Xi'an 710065, China; aomgj_97@163.com
*   Correspondence: huangchuang@nwpu.edu.cn; Tel.: +86-182-9199-9097

**Abstract:** High-speed supercavitating projectiles receive tremendous hydrodynamic force when flying underwater in tail-slap mode, and have obvious structural deformation and structural vibration. To study the motion characteristics of high-speed supercavitating projectiles, a bidirectional fluid-structure interaction model was established, and validated by comparing with the existing results. The motion, supercavitation flow field, and structural deformation response process of a supercavitating projectile were numerically investigated under the conditions of initial speed within 800–1600 m/s. It was found that the tail-slap motion of high-speed supercavitating projectiles is correlated with a high-frequency structural vibration. Further, the amplitude of the structural vibration increases with the initial speed. When flying with an initial speed higher than 1200 m/s, supercavitating projectiles encounter a great structural deformation under the action of the huge hydrodynamic load, which exerts a significant influence on the motion characteristic, and even destroys the trajectory stability. Thus, the supercavitating projectile cannot be regarded as a rigid body any more, and the structural response effect must be considered.

**Keywords:** structural response effect; high-speed supercavitating projectiles; tail-slap motion; motion characteristics; bidirectional fluid-structure interaction; computational fluid dynamic

## 1. Introduction

Supercavitating projectiles are kinetic energy weapons; they are launched by artillery and fly without any impetus. Using natural supercavitation drag reduction technology, supercavitating projectiles can fly underwater at a very high speed, and become an effective means for large combatant vessel against small underwater threats such as torpedoes, mines, and frogmen [1,2]. To extend the defense range and enhance power, supercavitating projectiles are expected to fly farther underwater and have greater residual kinetic energy at the end of trajectory. Increasing the initial speed is the most direct and effective technical means to achieve this target [3]. However, as the initial speed increases, the hydrodynamic load on supercavitating projectiles increases in a squared relationship. This causes significant structural deformation and even structural damage [4,5].

Using artillery firing technology, the initial firing speed of a 30 mm caliber artillery weapon can easily exceed 1700 m/s, which is higher than the sound speed in water [6]. Thus, the technical bottlenecks restricting the further increase of the initial speed of supercavitating projectiles lie in the structural vibration and structural strength. Hrubes [7] conducted a series of water entry experiments on high-speed supercavitating projectiles. It was found that the poorly designed projectiles would bend and deform when entering water with a high speed and small attitude angle, which eventually leads to the trajectory losing stability. Ishchenko et al. [8] shot supercavitating projectiles into water at an initial speed of nearly 1000 m/s, and analyzed the structural deformation phenomenon caused by the huge hydrodynamic force. Akbari et al. [9] studied the hydrodynamic characteristics of

supercavitating projectiles launched obliquely into water, and found that the supercavitating projectiles were subjected to a significant normal impact force which was closely related to the wet area. The experimental results of Chen et al. [10] showed that supercavitating projectiles could fly stably in a tail-slap mode, indicating that an approximately periodic hydrodynamic force acts on the supercavitating projectile during the underwater motion process. The alternating load can result in a high-frequency structural vibration, and even a sympathetic vibration, which exerts significant influence on the flow field in high-speed conditions. Moreover, a strong interaction effect exists among the structural deformation, flow field and hydrodynamic force, which is further enhanced with the increase of the initial speed.

Numerous results show that the fluid-structure interaction (FSI) effect must be considered when studying the hydrodynamic and motion characteristics of high-speed underwater moving bodies [11]. Zhou et al. [12] calculated the deformation of high-speed supercavitating projectiles by the modal superposition method. A method used to calculate the trajectory of nonrigid projectiles was also proposed, and the motion characteristics of the projectiles were analyzed by considering the rigid–flexible coupling effect. Chen et al. [13] established the FSI simulation model by utilizing a smoothed particle hydrodynamics method and improved boundary treatment. Wang et al. [14] numerically investigated of the tail-slap characteristics of high-speed supercavitating projectiles by using the Euler–Lagrange coupling method, and found that the angle of attack when touching the water surface is a vital factor affecting the trajectory stability. However, the supercavitating projectile is still treated as a rigid body and the structural deformation and FSI effect are not taken into account. Hu et al. [15] established an FSI model for a supercavitating projectile entering water vertically by using the arbitrary Lagrange–Euler method, and found that the maximum stress at the cavitator exceeds the limit stress of a supercavitating projectile made of tungsten alloy when the initial speed is more than 500 m/s. Therefore, it is believed that supercavitating projectiles lose rigidity gradually and show an elastic feature when the speed is further increased.

During the water-entry process, supercavitating projectiles are at the highest speed and receive the strongest impact load, and are most likely to lose structure and motion stability. However, if the posture and attack of the angle is suitable, the supercavitating projectile can enter the water surface safely and smoothly. Then, the supercavitating projectile flies underwater in a tail-slap motion mode. Wang et al. [16] developed a bidirectional FSI numerical method by solving the Reynolds average Navier–Stokes (RANS) equations and the structural dynamics equation simultaneously. This method can be used to calculate the structural deformation and hydrodynamic force of a flexible wing during the flapping process. Wu et al. [17] presented a high-fidelity numerical method to predict the unsteady interaction between fluid and structure, which can accurately predict the structural deformation response characteristics caused by a periodic hydrodynamic load. The alternating hydrodynamic force acting on the supercavitating projectile may result in a significant structural vibration, and even overturns the motion stability.

To achieve a farther underwater range, the launched speed of supercavitating projectiles is expected to be higher. However, a supercavitating projectile operating in high-speed conditions encounters new challenges because the material is not strong enough to support the huge hydrodynamic force. Under such conditions, significant structural vibration and structural deformation take place, which results in a change of the relative position between the supercavity and the supercavitating projectile. Conversely, the hydrodynamic force is determined by the relative position, and determines the deformation extent of the supercavitating projectile. In other words, there is a significant interaction between the structural deformation and the supercavitation flow field for high-speed supercavitating projectiles. The trajectory of supercavitating projectiles shows obvious tail-slap phenomenon, so the formed hydrodynamic force is quasi-periodic. The resulting structural response of supercavitating projectiles is totally different from that caused by the instantaneous impact load in the water-entry process.

However, research on the structural deformation and motion characteristics of high-speed supercavitating projectiles under an alternating load is very rare. Most of the studies about the hydrodynamic force and motion features of supercavitating projectiles are based on the rigid-body assumption. Thus, the achieved conclusions are not necessarily applicable for high-speed supercavitating projectiles. Some researchers have studied the water-entry process of low-speed supercavitating projectiles and focus on the structural deformation caused by the impact load, which provides approaches for studying the FSI effect of high-speed supercavitating projectiles. To reveal the structural response characteristics of high-speed supercavitating projectiles during the tail-slap motion, a bidirectional FSI method was utilized to model the flow field and structural deformation. In this method, the projectile motion, supercavitation flow, and structural response are solved simultaneously and the interactions among them are considered. The motion characteristics and structural deformation characteristics of the supercavitating projectile with initial speed within 800–1600 m/s are numerically investigated. The results can provide references for the structural design and engineering application of high-speed supercavitating projectiles.

## 2. Numerical Method

The method of computational fluid dynamics (CFD) is used to model the supercavitation flow field, and the method of computational structural dynamics (CSD) is applied to solve the structural deformation of the supercavitating projectile. The FSI simulation is achieved by constructing a bidirectional data transfer channel between the CFD solver and CSD solver.

### 2.1. Numerical Model of Supercavitation Flow Field

The external flow field of supercavitating projectiles includes two-phase flow and the mass transfer between the vapor phase and water phase, and a clear and stable interface exists between the two phases. The volume of fluid (VOF) model is used to solve multiphase flow fields, and is suitable for the flow field of which the interactions between different phases are not very strong and the interface is continuous and stable [18]. The VOF model solves only one set of momentum equations and calculates the multiphase flow by tracking the volume fraction of each phase, and is featured by high computational efficiency and strong adaptability for dealing with complex flow problems. Considering that the evaporation rate in the natural supercavitation problem is limited, and both the convective heat transfer coefficient and specific heat capacity of water are very high, no obvious temperature change occurs in the flow field. Moreover, the temperature is assumed to be a constant, equal to the room temperature (~15 °C), and the energy equation is not taken into account in modeling the flow field. Then, the basic control equations consist of the mass conservation equation and the momentum conservation equation.

For the $q$th phase of multiphase flow, the mass conservation equation is written as follows.

$$\frac{\partial}{\partial t}\left(\alpha_q \rho_q\right) + \nabla \cdot \left(\alpha_q \rho_q \vec{v}\right) = \sum_{p=1}^{n} \left(\dot{m}_{pq} - \dot{m}_{qp}\right) \tag{1}$$

where $\alpha_q$ is the volume fraction of the $q$th phase; $\vec{v}$ denotes the velocity, m/s; $\rho_q$ is the density of the $q$th phase, kg/m$^3$; $\dot{m}_{pq}$ and $\dot{m}_{qp}$ represent the mass transfer rate between the $p$th phase and the $q$th phase, kg/(m$^3$·s); $n$ is the total number of all the phases, and $n = 2$ for the supercavitation flow field.

To close the mass conservation equation, a relationship describing the sum of the volume fraction of each phase is supplemented as follows.

$$\sum_{q=1}^{n} \alpha_q = 1 \tag{2}$$

In a control volume, the momentum conservation equation based on the average density and the average kinetic viscosity can be described as follows.

$$\frac{\partial}{\partial t}\left(\rho_{\mathrm{m}}\vec{v}\right) + \nabla \cdot \left(\rho_{\mathrm{m}}\vec{v}\vec{v}\right) = -\nabla p + \nabla \cdot \left[\mu_{\mathrm{m}}\left(\nabla\vec{v} + \nabla\vec{v}^{T}\right)\right] + \rho_{\mathrm{m}}\vec{f} \tag{3}$$

where $\rho_{\mathrm{m}}$ is the average density of all the phases, kg/m$^3$; $\mu_{\mathrm{m}}$ is the average kinetic viscosity of all the phases, N·s/m$^2$. $\rho_{\mathrm{m}}$ and $\mu_{\mathrm{m}}$ can be expressed as follows.

$$\rho_{\mathrm{m}} = \sum_{p=1}^{n} \rho_p \alpha_p \tag{4}$$

$$\mu_{\mathrm{m}} = \sum_{p=1}^{n} \mu_p \alpha_p \tag{5}$$

The interphase mass transfer rate in the mass conservation equation can be calculated by a cavitation model. The Schnerr–Sauer cavitation model is derived from the Rayleigh–Plesset Equation, and the nucleate boiling regime is used to describe the evaporation process. It has the advantages of simple form, high computational efficiency [19], and strong numerical stability, and is expressed as follows.

$$\begin{cases} \dot{m}_e = \frac{\rho_v \rho_l}{\rho_m}\alpha(1-\alpha)\frac{3}{R_B}\sqrt{\frac{2}{3}\frac{P_v - P}{\rho_l}}, & P_v \geq P \\ \dot{m}_c = \frac{\rho_v \rho_l}{\rho_m}\alpha(1-\alpha)\frac{3}{R_B}\sqrt{\frac{2}{3}\frac{P - P_v}{\rho_l}}, & P_v < P \end{cases} \tag{6}$$

where $\dot{m}_e$ is the evaporation rate, kg/(m$^3$·s); $\dot{m}_c$ is the condensation rate, kg/(m$^3$·s); $P_v$ is the saturated vapor pressure at room temperature, Pa; $P$ is the local static pressure, Pa; $\rho_v$, $\rho_l$ and $\rho_m$ are the densities of the vapor phase, water phase, and mixture phase, respectively, kg/m$^3$; $\alpha$ denotes the volume fraction of the vapor phase; $R_B$ is the radius of the micro-bubbles, and is related to the quantity of microbubbles in per unit volume of water.

To close the control equations, a turbulence equation is included. The realizable *k-ε* turbulence model can achieve high simulation accuracy and strong numerical stability when used to calculate the supercavitation flow field which is adopted in this paper [20].

For a realizable k-epsilon turbulence model, the governing equations of turbulent kinetic energy and turbulent dissipation rate, and the model constants are expressed as follows.

$$\frac{\partial}{\partial t}(\rho k) + \nabla \cdot \left(\rho k \vec{v}\right) = \nabla \cdot \left[(\mu + \mu_t/\sigma_k)\nabla k\right] + P_k + G_b - \rho\varepsilon - Y_M \tag{7}$$

$$\frac{\partial}{\partial t}(\rho\varepsilon) + \nabla \cdot \left(\rho\varepsilon\vec{v}\right) = \nabla \cdot \left[(\mu + \mu_t/\sigma_k)\nabla\varepsilon\right] + \rho C_1 S\varepsilon - \frac{\rho C_2 \varepsilon^2}{k + \sqrt{v\varepsilon}} + C_{1\varepsilon}\frac{\varepsilon}{k}C_{3\varepsilon}G_b \tag{8}$$

where $C_1 = \max[0.43, \eta/\eta + 5]$, $\eta = Sk/\varepsilon$, $S = \sqrt{2S_{ij}S_{ji}}$, $P_k$ is the turbulent energy produced by the velocity gradient, $G_b$ is the turbulent energy produced by the velocity buoyancy, $Y_M$ is the correction term related to compressibility, and is equal to 0 for incompressible flow; $\mu_t = \rho C_\mu k^2/\varepsilon$ denotes the coefficient of turbulence viscosity; $C_\mu$ describes the relationship between average strain rate and rotation, and takes value of 0.09. $C_{1\varepsilon}$, $C_2$, $\sigma_k$, and $\sigma_\varepsilon$ are model constants, and take values of 1.44, 1.9, 1.0 and 1.2, respectively. $C_{3\varepsilon}$ is also a model constant, and takes the value of 1.0 in the gravity direction and 0 in the perpendicular direction.

However, the realizable *k-ε* turbulence model is only applicable to the fully developed turbulent flow field under high Reynolds number conditions. Thus, a wall function is also required to accurately calculate the wall shear stress. The scaled wall function is chosen as the supplement of the realizable *k-ε* turbulence model.

### 2.2. Numerical Model of Structural Deformation

A supercavitating projectile made of 93[#] tungsten alloy is adopted. Under high-speed conditions, the supercavitating projectile experiences large hydrodynamic forces, which causes structural deformation and structural vibration. The explicit dynamics approach is used to model the structural deformation process of the high-speed supercavitating projectile, characterized by a large strain rate and strong nonlinearity. To accurately calculate the transient structural deformation of supercavitating projectiles during the tail-slap motion process, a dynamic kinetics constitutive model is also required. The Johnson–Cook constitutive model can accurately describe the dynamic deformation characteristics of metallic materials in most cases [21], and is also suitable for computing the deformation and damage process of the supercavitating projectile made of 93[#] tungsten alloy. The Johnson–Cook constitutive model is expressed as follows.

$$\overline{\sigma} = \left[ A + B \left( \overline{\varepsilon}^{pl} \right)^{n} \right] \left[ 1 + C \ln \left( \frac{\dot{\overline{\varepsilon}}^{pl}}{\dot{\varepsilon}_0} \right) \right] \left( 1 - \hat{T}^{m} \right) \tag{9}$$

where $\hat{T} = (T - T_r)/(T_{melt} - T_r)$, $T_{melt}$ is the material melting point, takes value of, K; $A$, $B$, $C$, $m$ and $n$ are the model constants, and achieved by mechanical property test; $A$ is the quasi-static yield stress, MPa, $B$ is the strain hardening constant, MPa, $C$ is the strain rate correlation coefficient, $m$ is the temperature correlation coefficient, $n$ is the strain-hardening exponent. For 93[#] tungsten alloy, $T_{melt}$ = 1723 K, $A$ = 1306 MPa, $B$ = 450 MPa, $C$ = 0.016, $m$ = 1.00, and $n$ = 0.12.

In this study, the supercavitating projectiles fly underwater and are subjected to approximately no frictional drag, and the deformation process is very short, so the heat produced at the surface of the supercavitating projectile is very limited. In addition, the natural cavitation is a slightly endothermic process because a small quantity of water evaporates into vapor. Considering the quantity of water is huge and the convective heat transfer coefficient is very high, no obvious temperature difference exists between the solid and fluid. Thus, the temperature on the supercavitating projectile is also assumed to be a constant equal to the room temperature. Then, the item $\hat{T}$ in Equation (9) is zero. That is, the softening of material resulting from temperature increase is not included in modeling the structural dynamics of the supercavitating projectile.

### 2.3. Bidirectional Fluid Structure Interaction Simulation Method

For high-speed supercavitating projectiles, a strong interaction exists between the flow field and the structural deformation; this is a typical bidirectional FSI problem. The data from the CFD solver and CSD solver are required to be transferred and exchanged mutually on the coupling surface.

In this paper, STAR-CCM+ 13.04 was chosen as the CFD solver, while Abaqus 6.14 was selected as the CSD solver. The co-simulation of the flow field and structural deformation was achieved by using a segregated solution strategy. The control equations of the flow field and structure were solved separately using the solvers. A bidirectional FSI simulation was achieved by setting a coupling surface, of which the data from the two solvers could transfer from one to the other smoothly and in an orderly manner. The solution procedure of the bidirectional FSI simulation is shown in Figure 1, and is summarized as follows.

*Step 1*: The external flow field of the supercavitating projectile is calculated by the CFD solver, and the instantaneous hydrodynamic force and the pressure distribution on the coupling surface are achieved.

*Step 2*: The pressure distribution is transferred to the CSD solver, and the instantaneous deformation of the supercavitating projectile is calculated.

*Step 3*: The local deformation of each node on the coupling surface is obtained from the CSD result.

***Step 4***: The deformed supercavitating projectile is adopted as the geometric model to update the grid nodes of the flow field, and the next timestep begins by going back to Step 1.

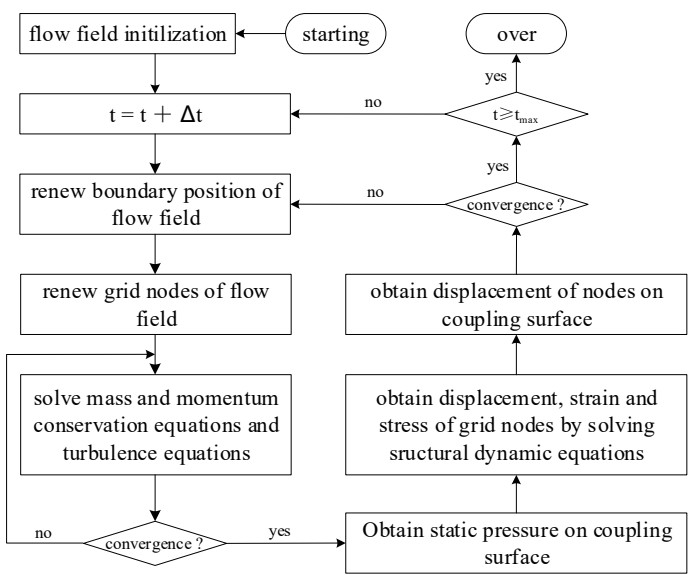

**Figure 1.** Flow chart of bidirectional FSI simulation.

*2.4. Computation Domain and Boundary Conditions*

A 30 mm caliber supercavitating projectile was investigated in this paper. The outline and key parameters are shown in Figure 2. The supercavitating projectile consisted of a cylinder section and a frustum section, and the whole length was 240 mm. The diameter ($D_c$) and length ($L$) of the cylinder section were 30 mm and 36 mm, respectively. The larger end-face of the frustum section was linked with the cylinder section, the smaller end-face operated as a disk cavitator, and the diameter was 6 mm. Additionally, the supercavitating projectile was made of 93# tungsten alloy, the mass was 1.28 kg, and the mass center was 165 mm away from the cavitator.

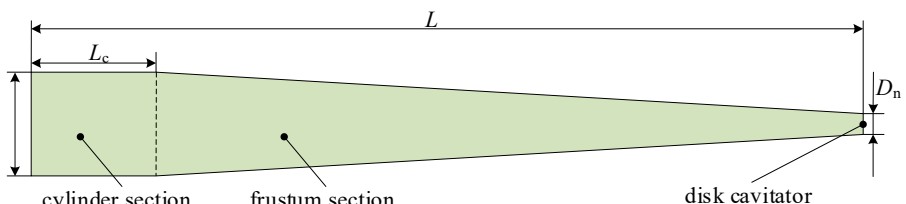

**Figure 2.** Layout and geometric parameters of supercavitating projectile.

When flying at a 1 m depth underwater with a speed of 1000 m/s, the whole length of the produced supercavity can be 150 times longer than that of the supercavitating projectile. However, only the part of supercavity near the supercavitating projectile exerts influence on the hydrodynamic force motion characteristics [22]. Therefore, it was not necessary to calculate the complete supercavity by establishing a very large computation domain. To save computational cost, only the flow field near the supercavitating projectile was modeled in this paper. A fluid domain and a solid domain were included in the computational domain. The solid domain denotes the supercavitating projectile, and the fluid domain is the surrounding flow field. By comparison and analysis, the ultimately selected fluid domain had the shape of cylinder, and the diameter was 50 $D_c$. The full length is 11 $L$. This is shown in Figure 3. The inlet and outlet of the computational domain were 4 $L$ and 7 $L$ away from the cavitator, respectively. In addition, the left side and the cylindrical surface of the computational domain were regarded as a pressure inlet boundary condition, and the

total pressure of the incoming flow was set to 111,125 Pa, which is the hydrostatic pressure at the depth of 1 m underwater. The right side was a pressure outlet, and the static pressure was also set as 111,125 Pa. The walls of the supercavitating projectile were the coupling surface between the fluid and solid domains, and were set as a non-slipping wall.

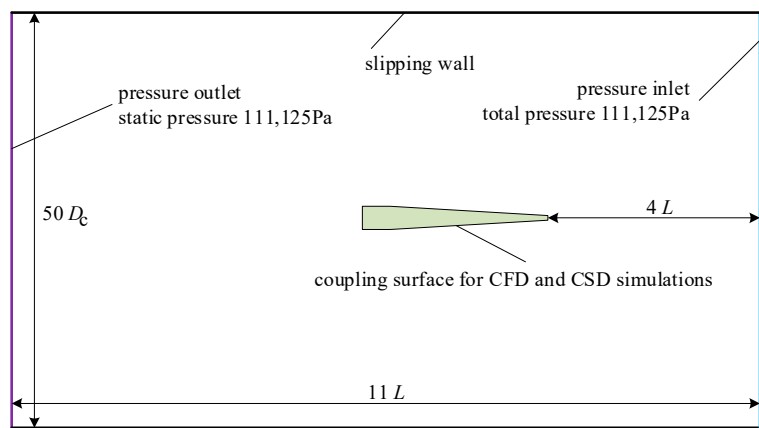

**Figure 3.** Schematic diagram of computational domain and boundary conditions.

Both the fluid and solid domains were partitioned by using the trimmed Mesher. In the region near the interface between water and vapor, stagnation point, and wake, the flow field was more complex, and gradients of pressure, velocity, and other flow parameters were always greater or changed drastically. To achieve a convergent and accurate calculating result, the grids in these regions were well refined. In addition, prismatic layer grids were arranged in the region near the surface of the supercavitating projectile, which ensured the precise simulation of the flow field in the boundary layer. A practicable meshing result of the computational domain was finally achieved by optimizing the distribution of grid nodes again and again, and this is shown in Figure 4. To verify the grid independence, three cases with different nodes were designed. Case 1: the grid nodes in the fluid domain and solid domain were 0.61 million and 0.05 million, respectively; Case 2: the grid nodes in the fluid domain and solid domain were 1.23 million and 0.11 million, respectively; Case 3: the grid nodes in the fluid domain and solid domain were 2.45 million and 0.22 million, respectively. By comparison and analysis, the grids satisfied the requirement of grid independence, and this work is not presented in this paper.

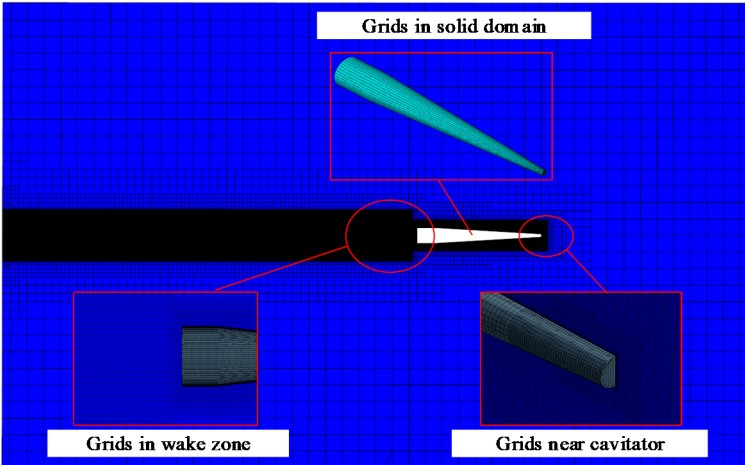

**Figure 4.** Distribution of grids in fluid and solid domain.

### 3. Model Validation

The issues of phase change, unsteady flow, motion, and FSI effect were included in the simulation of the structural deformation response in a supercavitation flow field. No published experimental results exactly addressing this problem have been found. To validate the numerical model, the supercavity shape, trajectory characteristic, and dynamic structural deformation law in flow field were simulated, and compared with the published conclusions. The calculated results of the supercavity shape and trajectory characteristic of a rigid supercavitating projectile without considering the FSI effect are well verified and have sufficient precision, and are presented in a previous work [22]. To validate applicability for calculating the structural deformation response process, a published result about the FSI effect in a single-phase flow field is adopted in this paper.

Walhorn et al. [23] obtained the vortex-induced vibration process of a flexible plate put behind a square column in the water, and this can be utilized as a case to validate the numerical model for the FSI problem. According to the geometric parameters, physical properties, and operating conditions presented in Ref. [23], the numerical model built in Section 2 was used to simulate the FSI process. The dynamic structural deformation of the flexible flat was obtained. When the simulation time was equal to 4.0 s, the calculated results of the flow field and structural deformation were extracted and compared with the published results. The comparison is shown in Figure 5.

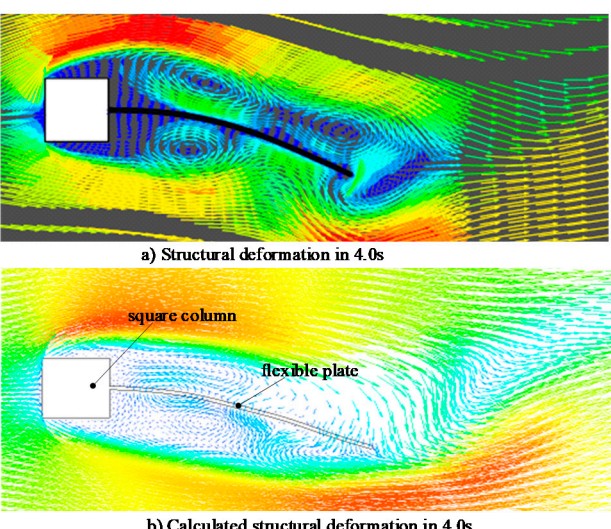

a) Structural deformation in 4.0s

square column

flexible plate

b) Calculated structural deformation in 4.0s

**Figure 5.** Comparison of calculated instantaneous deformation of flexible plate at *t* = 4.0 s with result in Ref. [23].

According to the calculation results, the location of the end of the flexible plate at different times was extracted, and compared with the one in Ref. [23] as shown in Figure 6. The obtained period of the vibration at the end was 0.311 s, which is 5.76% longer than the result in Ref. [23]. The calculated average amplitude at the end was 963.4 mm, which is 5.41% larger than the result in Ref. [23]. Therefore, The FSI numerical model established in this paper can accurately predict the dynamic structural deformation characteristics in an unsteady flow field.

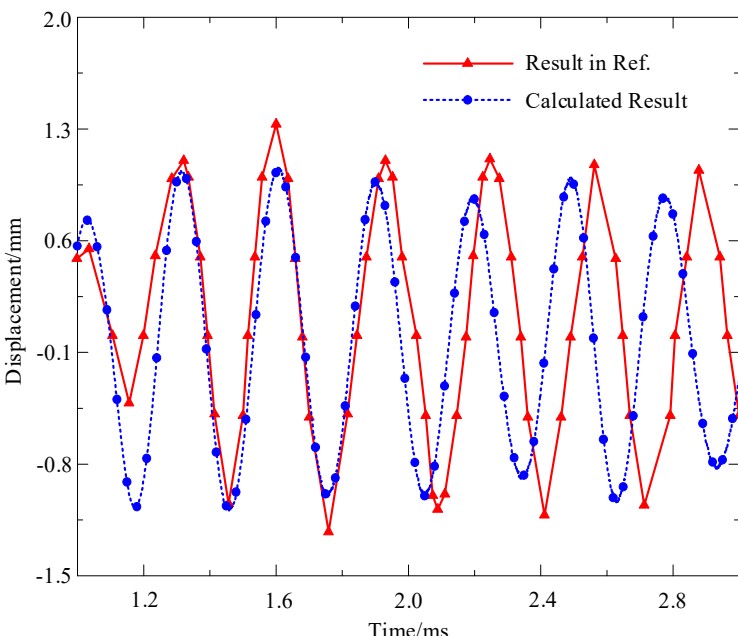

**Figure 6.** Comparison of calculated vibration feature of plate with result in Ref. [23].

## 4. Results and Discussion

The magnitude of the structural deformation of the supercavitating projectile is positively related to the hydrodynamic force, which is proportional the square of the speed. Therefore, no significant deformation is found when a supercavitating projectile is flying at a low speed. Supercavitating projectiles with an initial speed in the range of 800–1600 m/s were chosen to perform the FSI numerical simulation, and to study the hydrodynamic characteristics and the structural deformation response characteristics by considering the FSI effect. The interval of the calculated initial speed was 200 m/s, and bidirectional FSI effect numerical simulations were performed for the operating conditions with initial speed of 800, 1000, 120,1400 and 1600 m/s.

### 4.1. FSI Simulation Results of Supercavitating Projectile

From the FSI simulation results, the flow parameters, supercavity shape, hydrodynamic force acting on the supercavitating projectile, motion characteristics of the supercavitating projectile, and dynamic deformation response process of the supercavitating projectile were obtained. Taking the case with an initial velocity of 1600 m/s as an example, the motion characteristics and structural deformation characteristics are presented. The tail-slap motion of the supercavitating projectile was gradually formed under the action of the initial perturbance of initial angle of attack or angular velocity. The tail-slap motion could not be unceasingly maintained and the stability of the trajectory was suddenly broken, which was mainly caused by the excessive structural deformation of the supercavitating projectile. The displacement of the cavitator center relative to the mass center according to the FSI simulation results was extracted and is displayed in Figure 7. From the supercavitating projectile initially released to the moment of finally losing the trajectory stability, the structural deformation response underwent three typical stages: initial vibration, amplitude amplification, and vibration divergence.

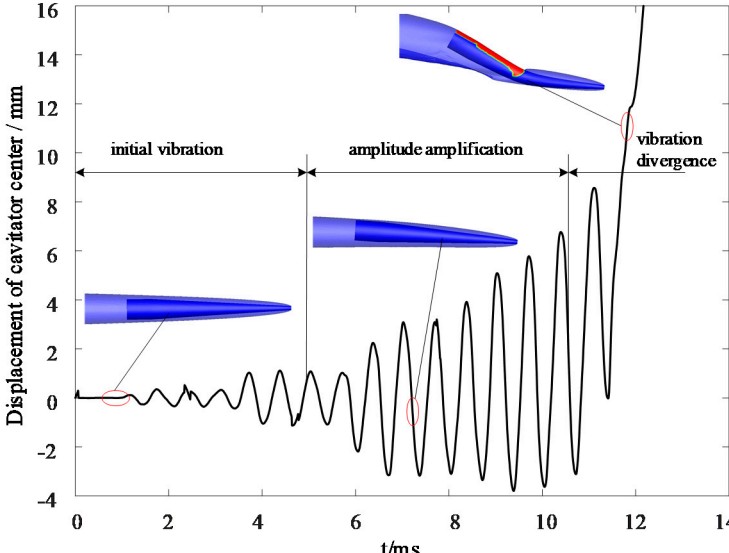

**Figure 7.** Vibration characteristics of cavitator center by considering FSI effect (1600 m/s).

After entering into water, the supercavitating projectile flies and is accompanied by a supercavity. Under the action of initial perturbance, the supercavitating projectile rotates slowly around the mass center, which results in a gradual increase in the angle of attack, and the side surface begins to touch the water by puncturing the supercavity. Further, the supercavitating projectile receives a righting moment, and rotates to the reverse direction and punctures the other side of the supercavity. Then, a continuous tail-slap motion is gradually formed. In this stage, the relative position between the supercavitating projectile and supercavity are displayed in Figure 8. The amplitude of tail-slap motion is limited, and the relative offset between the supercavitating projectile and supercavity is very small. Thus, the produced hydrodynamic force is not strong enough to cause a significant structural deformation. In this condition, the FSI effect is very weak, and the supercavitating projectile exhibits a feature like a rigid body.

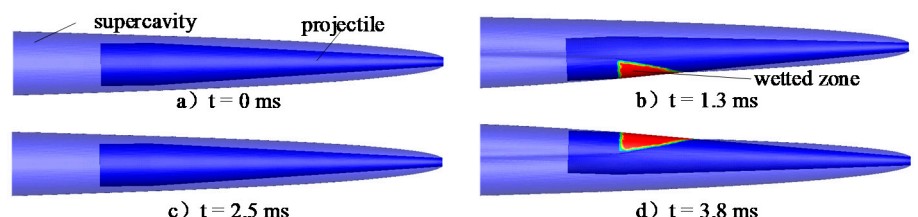

**Figure 8.** Relative position between supercavitating projectile and supercavity at the initial stage.

Subsequently, the amplitude of the tail-slap motion of the supercavitating projectile is enlarged significantly. Resultantly, the effective angle of attack increases synchronously, as well as the hydrodynamic force. Then, the strong alternating force leads to a high frequency, and a large structural deformation response is subsequently observed, and the amplitude of the structural vibration is gradually enlarged. This is the amplitude amplification stage, as shown in Figure 7. In this stage, the maximum stress of the supercavitating projectile is no more than the yield limit of 93# tungsten alloy, and the structural deformation is categorized to elastic deformation. Because of the structural deformation, the axis of the supercavitating projectile bends significantly during the tail-slap motion. Then, the wetted status is not only determined by the relative position between the supercavitating projectile and the supercavity, but also changed with the bending condition of the axis of the supercavitating projectile. Resultantly, the wetted area is somewhat further increased or decreased, which is determined by the frequencies and phase position of the tail-slap motion and structural vibration.

The deformation status of the supercavitating projectile and the relative position between the supercavitating projectile and the supercavity in the amplitude amplification stage are exhibited in Figure 9. When the simulation time is 6.0 ms, the supercavitating projectile axis is upward-bending. Though the tail of supercavitating projectile punctures the supercavity and touches water at the lower side, the wetted area is small. In this condition, the offset directions of the supercavitating projectile tail caused by the structural deformation and tail-slap motion are opposite, and can be partially cancelled out. Therefore, the structural deformation decreases the wetted area and the hydrodynamic force. At 6.5 ms, the axis of the supercavitating projectile turns to concave-bending. The cavitator center shifts upward, and the axis of the produced supercavity locates above the axis of the supercavitating projectile. Therefore, a larger wetted zone at the lower side of the tail of supercavitating projectile occurs when puncturing the supercavity. When the angle of attack is positive and the lower side of the supercavitating projectile is wetted, the concave-bending deformation increases the wetted area, which is induced under a greater hydrodynamic force. Resultantly, the structural deformation further increases, and positive feedback takes place.

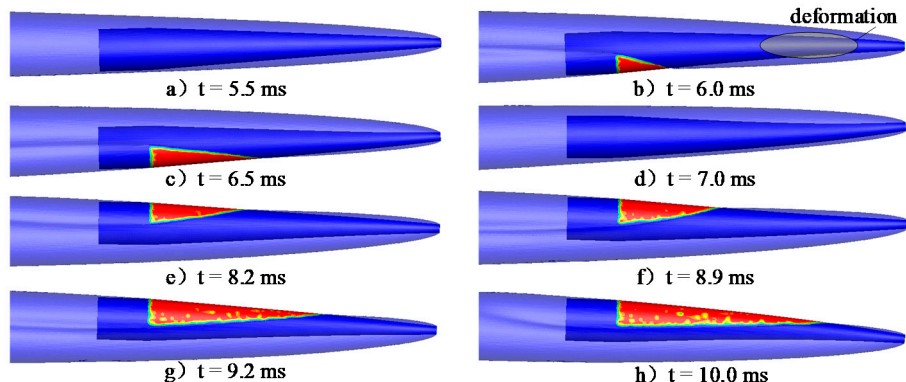

**Figure 9.** Deformation of supercavitating projectile and wetted status in amplitude amplification stage.

In the period of 8.2–10 ms, both the wetted area and the structural deformation of the supercavitating projectile continuously increase, and the supercavity shows an obvious asymmetric shape due to the significant changes in both the position and axial direction of the cavitator. In this stage, the structural deformation of the supercavitating projectile is significant, and exerts great influence on the position and shape of the supercavity. Moreover, the changes in flow field, in turn, affect the structural deformation through changing the hydrodynamic force. Therefore, the supercavitating projectile cannot be regarded as a rigid body because of the strong bidirectional FSI effect.

When the simulation time is equal to 10.6 ms, the supercavitating projectile comes into the vibration divergence stage. The deformation of the supercavitating projectile and the produced supercavity at this stage are displayed in Figure 10. In this stage, both the amplitudes of the structural vibration and the tail-slap motion suddenly increase by a large margin in a short period of time, and sharply lose stability. At 10.6 ms, the deformation of the supercavitating projectile is still not very large, the axis also looks like a line, and the supercavity can fully envelop the supercavitating projectile.

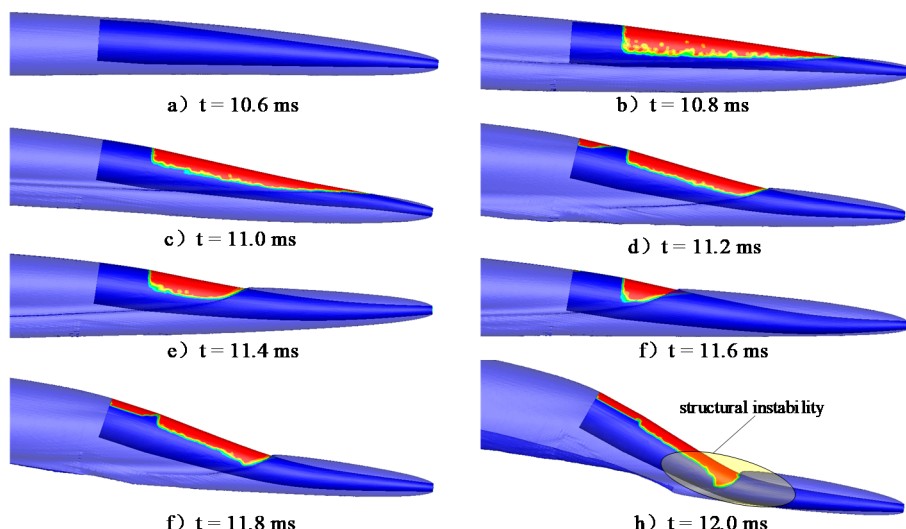

**Figure 10.** Deformation of supercavitating projectile and wetted status in vibration divergence stage.

However, at 10.8 ms, the supercavitating projectile axis becomes S-shaped, the cavitator center is below the mass center, and the tail center is above the mass center. Therefore, a very large offset between the supercavity and supercavitating projectile is produced, which leads to the supercavitating projectile being suddenly wetted in a large area, and subjected to a huge hydrodynamic force. Resultantly, there is a greater concave bending of the supercavitating projectile. Then, the motion stability, the continuity of the supercavity, and the restorability of the structural deformation further deteriorate, and the coupling effect of the structural vibration and the tail-slap motion promote this process. Finally, an unrecoverable structural deformation is formed in a short period of time, and directly causes the instability of the underwater trajectory. This is regarded as the main reason why the structure and motion of the high-speed projectile suddenly become unstable during the tail-slap motion underwater.

*4.2. Influence of FSI Effect on Motion Characteristics of Supercavitating Projectile*

The characteristics of the hydrodynamic force acting on the supercavitating projectile during the tail-slap motion are also obtained by considering the FSI effect. For the cases with initial speeds of 800, 1000, 1200, 1400, and 1600 m/s, the simulations on the characteristics of motion and hydrodynamic force of a rigid supercavitating projectile are also performed as a comparison to distinguish the influence of the FSI effect. The resulting hydrodynamic forces for supercavitating projectiles flying at different initial speeds are compared in Figure 11. The structural deformation is directly represented by the magnitude of the hydrodynamic force. The dimensional form of the hydrodynamic forces of different cases is displayed in Figure 11. Then, to exhibit the difference clearly, a comparison of cases with initial speeds of 800 m/s and 1000 m/s is displayed in Figure 11a, and a comparison of cases with initial speeds of 1200 m/s, 1400 m/s, and 1600 m/s is shown in Figure 11b.

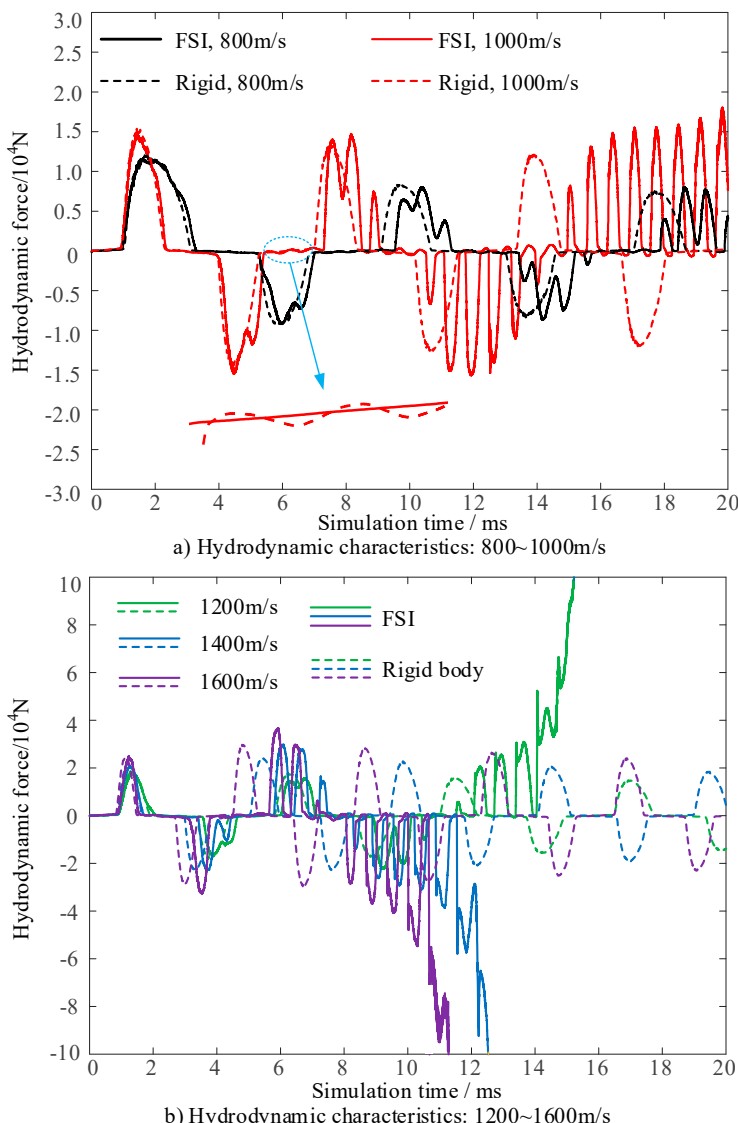

**Figure 11.** Influence of FSI effect on hydrodynamic characteristics under different initial speeds. (**a**) Hydrodynamic characteristics: 800~1000 m/s (**b**) Hydrodynamic characteristics: 1200~1600 m/s.

As shown in Figure 11a,b, the presented hydrodynamic characteristics during the tail-slap motion of the supercavitating projectile while considering the FSI effect are very different from those of the rigid supercavitating projectile. For the cases with initial speeds of 800 m/s and 100 m/s, the hydrodynamic force fluctuates at a low frequency because the upper side and the lower side of the supercavitating projectile touch water alternately during the tail-slap motion. Moreover, a high frequency and small amplitude fluctuation is observed in the hydrodynamic force of the supercavitating projectile, which is caused by the structural vibration. Because the frequency of the structural vibration is much higher than that of the tail-slap motion, the wetted status of the supercavitating projectile can change many times in one tail-slap period. Within the initial 20 ms, both the tail-slap motion and the structural vibration are stable. Thus, the supercavitating projectile can keep moving forward. In addition, for cases with an initial speed of 800 m/s, the hydrodynamic characteristics of the supercavitating projectile, considering the FSI effect, are similar to those of the rigid body. Thus, the FSI effect is not very strong, and can be neglected for simplification.

As shown in Figure 11a, the hydrodynamic characteristics for the case of 1000 m/s are scaled up within 6.0–8.0 ms by comparing them with the case of 800 m/s. It is found that the

high-frequency fluctuation of the hydrodynamic force remains when the supercavitating projectile is fully enveloped by the supercavity. The structural vibration does not disappear, even though the supercavitating projectile is fully covered by the supercavity and no more wetted. However, the position and direction of the cavitator changes continuously at a very high frequency, which causes a small amplitude and high-frequency hydrodynamic force acting on the supercavitating projectile. If the FSI effect is not taken into account, this phenomenon cannot happen.

With the increase of initial speed, the FSI effect becomes significant. For the case with the initial speed of 1200 m/s, the hydrodynamic characteristics of the supercavitating projectile, considering the FSI effect, are totally different from those of the rigid body. By considering the FSI effect, the motion and the structural vibration suddenly become unstable at about 2–3 times of the tail-slap period after being released. Furter, the higher the initial speed is, the earlier the supercavitating projectile loses stability. In these high-speed conditions, the hydrodynamic force is strong enough to cause the obvious structural vibration, which thoroughly changes the surrounding flow field and the motion characteristics. The unrecoverable structural deformation directly causes the trajectory to lose stability.

To further explain the suddenly loss of stability of the tail-slap motion of the supercavitating projectile, the evolution processes of the structural vibration of cases with different initial speeds are compared in Figure 12. When the structural vibration takes place, the cavitator center swings relative to the mass center at a high frequency. Then, the displacement of the cavitator center changes at a high frequency, and the amplitude and period are used to represent the structural vibration state. At the released time ($t = 0$ ms), no initial vibration is performed to the supercavitating projectile, and it just rotates around the mass center with an initial angular speed. According to Figures 9 and 11, the angle of attack increases gradually because of the initial rotation, and the tail-slap motion is subsequently formed. Then, the structural vibration is gradually formed under the action of the hydrodynamic force coming from the tail-slap motion, and the amplitude increases with the calculation time.

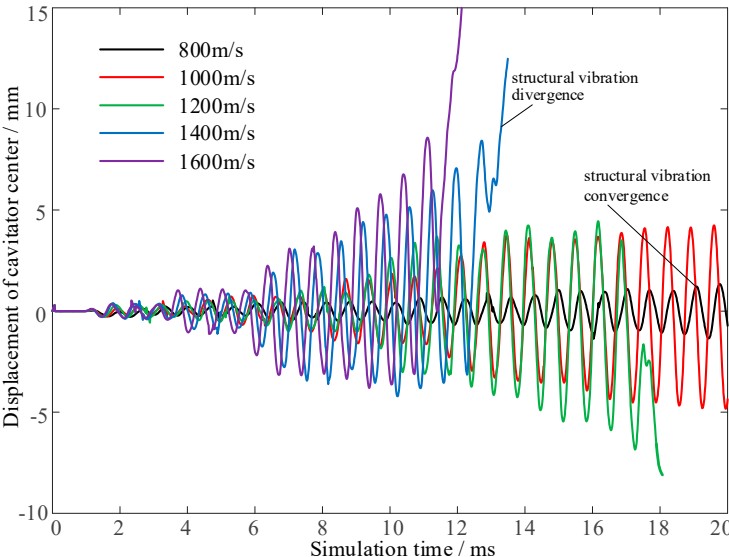

**Figure 12.** Influence of FSI effect on structural vibration characteristics under different initial speeds.

As displayed in Figure 12, the structural vibration can be continuous for cases with initial speeds of 800 and 1000 m/s, and the amplitudes converge to an approximately unchanged value. In addition, the convergent value of the amplitude increases with the initial value. For cases with initial speeds of 1200, 1400, and 1600 m/s, the overlarge amplitude of structural vibration results in an unrecoverable structural deformation, and causes the divergence of the structural vibration. It is worth noting that cases with a higher initial speed will diverge in a shorter time. Further, the frequency of the structural vibration

is very high, and is about 10 times higher than that of the tail-slap motion. The frequency of the tail-slap motion corresponds to that of the hydrodynamic force. Then, the structural vibration is formed under the alternating hydrodynamic force. The shown frequency is mainly determined by the inherent characteristics, and is not significantly influenced by the initial speed.

To further analyze the structural vibration characteristics of the supercavitating projectile during the tail-slap motion, the Fast Fourier Transform is performed. Then, the frequency domain characteristic of the elastic and rigid supercavitating projectiles under different initial speed conditions are obtained and compared. The comparison of the case with an initial speed of 1600 m/s is taken as an example and displayed in Figure 13. For the case ignoring the FSI effect, only one basic frequency of about 200 Hz is found from the frequency spectrum of the hydrodynamic force, which is identical to the frequency of the tail-slap motion. For the case considering the FSI effect, two basic frequencies are observed in the frequency spectrum of the hydrodynamic force; one is 170 Hz, and the other is1469 Hz. The smaller one corresponds to the tail-slap motion, and the higher one is the first-order bending intrinsic frequency of the supercavitating projectile obtained by modal analysis. When considering the FSI effect, the frequency of the tail-slap motion slightly decreases. By considering the FSI effect, the supercavitating projectile is regarded as an elastic body. It is equivalent to adding a buffer between the supercavitating projectile and water when puncturing the supercavity. In addition, the low-frequency hydrodynamic force motivates the intrinsic modality of the supercavitating projectile, and a high frequency structural vibration is formed.

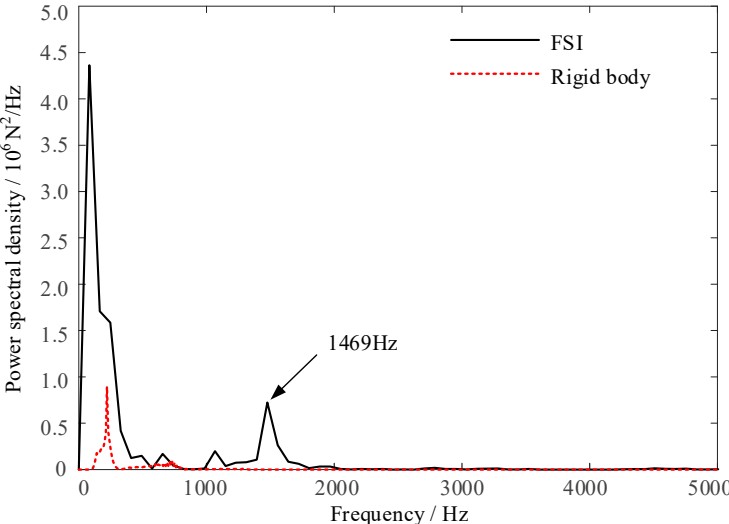

**Figure 13.** Frequency characteristics of the hydrodynamic force of elastic and rigid supercavitating projectiles.

According to the numerical results of the cases with different initial speeds, the changes of the pitch angle of the nonrigid and rigid supercavitating projectiles are extracted, respectively, and compared in Figure 14. When the initial speed is equal to 800 m/s, the change of the pitch angle along with time of the supercavitating projectile considering the FSI effect is approximately the same as that of the rigid one, and the tail-slap motion is stable. It is demonstrated that the FSI effect exerts less influence on the motion characteristics of the supercavitating projectile with an initial speed of 800 m/s, and does not break the stability of the trajectory.

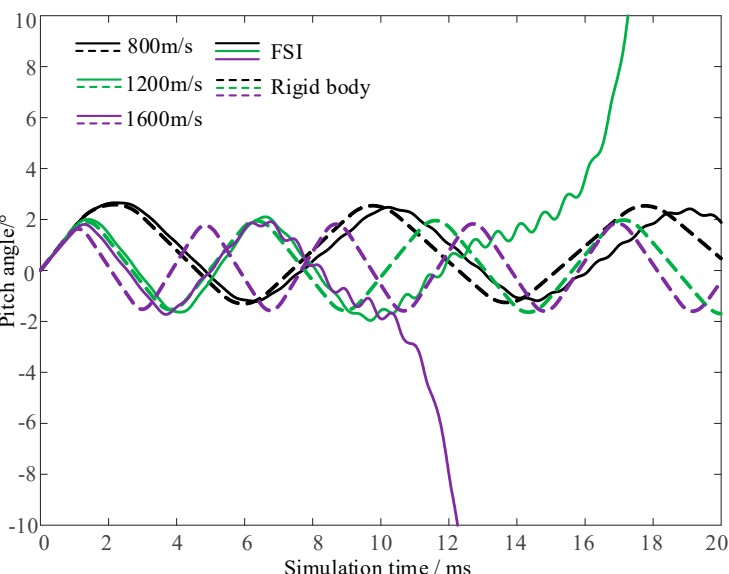

**Figure 14.** Changes of pitch angle of elastic and rigid supercavitating projectiles during tail-slap motion.

As the initial speed increases, the influence of the FSI effect becomes more and more significant, and results in the motion characteristics of the supercavitating projectile being obviously different from those of the rigid supercavitating projectile, and even causes the divergences of the tail-slap motion and structural vibration. For the case with an initial speed of 1200 m/s, the motion of the supercavitating projectile becomes unstable in about 15.0 ms. After being released, the tail-slap motion is formed step by step, and the amplitude of the angle of attack increases gradually. The deformation of the supercavitating projectile is positively related to the hydrodynamic force, and the hydrodynamic force is mainly determined by the speed and the angle of attack. For the high-speed cases, the threshold value of the hydrodynamic force to produce an unrecoverable deformation can be attained at a small angle of attack. However, for the low-speed cases, a large enough angle of attack is essential to reach to the threshold value. Therefore, the higher the initial speed is, the earlier the motion stability is destroyed.

During the tail-slap motion, the speed of the supercavitating projectile decreases continuously because of the drag force coming from the water. According to the calculation results of considering and ignoring the FSI effect, the speed decrease laws of supercavitating projectiles with different initial speeds are extracted, and these are compared in Figure 15. The value of each line at time zero denotes the initial speed. In cases both considering and ignoring the FSI effect, the supercavitating projectiles fly with a gradually decreasing speed, and no obvious difference is found between the change laws of speed of the two cases if the motion stability is not broken. For the calculated range of the speed and time, the FSI effect exerts less influence on the change law of the speed when the initial speed is no more than 1000 m/s. Moreover, if the initial speed exceeds 1200 m/s, the tail-slap motion and structural vibration of the supercavitating projectile cannot be maintained all the time because of the overlarge structural deformation. In this condition, the supercavitating projectile greatly deviates from the supercavity, and receives a large drag and upsetting moment. The upsetting moment leads to the further rotation of the supercavitating projectile, and a bigger angle of attack is formed. Resultantly, the drag increases in a positive feedback mode. Then, the speed decreases sharply when the motion stability is broken, and the supercavitating projectile is out of operation.

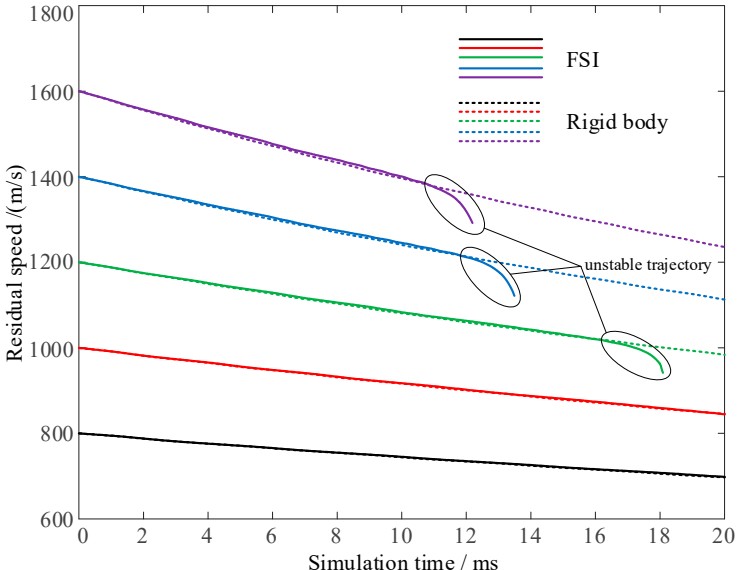

**Figure 15.** Change of speed of elastic and rigid supercavitating projectiles.

## 5. Conclusions

To study the influence of structural deformation response on the motion stability of high-speed supercavitating projectiles, a numerical model was established by considering the interaction between the supercavitation flow field and the structural deformation. Then, the hydrodynamic force characteristics, structural deformation response characteristics, and the underwater trajectory characteristics of the high-speed supercavitating projectile were numerically investigated, and the influence of the initial speed on the characteristics of the structural deformation and tail-slap motion were obtained. The key findings were as follows.

1. By combining the CFD method and CSD method, a bidirectional FSI method was developed to solve the coupling problem between the supercavitation flow field and the structural deformation of high-speed supercavitating projectiles. Then, the feasibility of the method was validated by comparison with the existing results.
2. As the initial speed increases, the structural deformation response of the supercavitating projectile gradually becomes significant, and exerts strong influence on the characteristics of the hydrodynamic force and tail-slap motion. Then, the rigid body assumption is gradually no longer invalid, and the FSI effect must be considered. If the initial speed exceeds the critical value, the trajectory stability can be broken because of the overlarge structural deformation. Moreover, the higher the initial velocity is, the faster the trajectory loses stability. In addition, the critical initial speed is about 1200 m/s for the researched supercavitating projectile.
3. For high-speed conditions, an obvious interaction exists between the structural vibration and the tail-slap motion. Two basic frequencies were found in the structural vibration caused by tail-slap motion. One is equal to the frequency of hydrodynamic force, and the other is approximately the first-order intrinsic frequency. The structural vibration causes a slight decrease of the frequency of the tail-slap motion in comparison with the results of rigid supercavitating projectiles.

**Author Contributions:** Conceptualization, C.H. (Chuang Huang) and K.L.; methodology, Z.L. (Zhao Liu) and C.H. (Changle Hao); software, C.H. (Changle Hao) and Z.L. (Zixian Liu); validation, D.L., Z.L. (Zhao Liu); formal analysis, K.L.; investigation, Z.L. (Zixian Liu); resources, C.H. (Chuang Huang); data curation, D.L.; writing—original draft preparation, C.H. (Changle Hao); writing—review and editing, Z.L. (Zhao Liu); visualization, K.L.; supervision, K.L.; project administration, D.L.; funding acquisition, C.H. (Chuang Huang). All authors have read and agreed to the published version of the manuscript.

**Funding:** This research was funded by the National Natural Science Foundation of China, grant number 51909218, China Postdoctoral Science Foundation, grant number 2019M653747.

**Institutional Review Board Statement:** Not applicable.

**Informed Consent Statement:** Not applicable.

**Conflicts of Interest:** The authors declare no conflict of interest.

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
