# Peer review of "Motion Characteristics of High-Speed Supercavitating Projectiles Including Structural Deformation"

_energies, doi:10.3390/en15051933_

Round 1
Reviewer 1 Report
A bidirectional fluid-structure interaction model was developed and validated by authors to study the motion characteristics of high-speed supercavitating projectiles. The motion, supercavitation flow field, and structural deformation response process of a supercavitating projectile were numerically investigated under the conditions of initial speed within 800~1600m/s. Overall, the paper is within the scope of Energies Journal. Authors are encouraged to address the following minor issues that need to be addressed before the consideration of paper.
-In abstract, its better to use “with the existing results.”
-Authors talk about convergent but the referee did not find convergence results. Please add
-Please include/expand scientific and physical interpretation for findings reported in some of the figures.
Author Response
Dear Reviewers, Thank you for the comments, they are helpful for me to perfect this manuscript (Manuscript ID: energies-1592501). I have revised this manuscript according your comments, the modifications are marked by using the “Track Changes” function of MS Word. In the attachment, the corresponding explanation and rebuttal are below each question, the questions are marked with blue, the answers are written with black. Hope my explanations can answer your questions. Please do not hesitate to contact me if you need further clarification. Chuang Huang March 1, 2022

Reviewer 2 Report
This paper deals with Motion Characteristics of High-speed Super-cavitation of Projectiles including fluid structural interaction (FSI). This area is quite challenging to tackle. There are a lot of disciplines involved in modeling super cavitation such as: heat transfer, two phase flow, thermodynamics, structural mechanics (including non-linear phenomena such as: thermal fatigue) and corrosion (pitting, stress corrosion cracking, cavitation erosion). Computational fluid dynamics (CFD) has been applied in order to model the super-cavitation flow field, and the method of computational structural dynamics (CSD) has been used in order to solve the structural deformation of the super-cavitation of projectile. The Fluid Structural Interaction (FSI) simulation is achieved by coupling the CFD solver and CSD solver. The authors have applied Johnson Cook constitutive model to describe the dynamic deformation of the metallic structure of the projectile. This Model is the most widely used pure empirical model for dynamic characterization and readily available in the numerical codes such as: ABAQUS, LsDyna etc. This model explicitly takes into account the hardening due to strain and strain rate, and also thermal softening due to high temperature caused by adiabatic heating. They have also validated their numerical results against the model obtained by Walhorn et al. [23]. The cavitation erosion phenomenon hasn’t been addressed in this research work. The energy transport equation (or heat transfer equation) is missing. The authors should add it with the boundary conditions to this manuscript. They should solve this equation in order to obtain the temperature profile at the interface of the gas/liquid phases. It should be noted Johnson cook model (Eq. 7) is based on the surface temperature of the projectile. The value of the surface temperature of the projectile is missing. To the best of my knowledge the authors should couple the numerical solution of the heat transfer equation in the bulk with the numerical solution of the heat conduction in the metallic surface (see my remark no 8). The values of the k-e constants, applied in the RANS turbulence model are missing and should be also described. It is recommended to consider this paper for publication, after performing the following major revisions.
Comments and Suggestions for Authors
1) The novelty of this study should be strengthen and emphasized based on gaps of current literature.
2) The authors should add more keywords after the abstract such as: CFD, FSI.
3) The introduction doesn’t explain the thermodynamic aspects of high-speed super-cavitation of projectiles, and this makes difficult for readers the comprehension of the novelty related to utilization of the super-cavitation technology.
4) (Introduction section) It is recommended to cite the following classical book in the introduction section:
Greiner, Leonard, Underwater Missile Propulsion, Compass Publications, Inc., Copyright 1967.
This book contains collection of papers presented at a symposium earlier in the 1960's and finally assembled in this now out-of-print book. It summarizes work done at Penn State University's Applied Research Laboratory.
5) (Section 2.1 line no. 117) the energy transport (heat transfer) equation is missing. The authors should add it with the boundary conditions to this manuscript. They should solve this equation in order to obtain the temperature profile at the interface of the gas/liquid phases. See the following paper:
Jianing T., Shipeng L., Ningfei W., Yingjie W. and Wei S.. "Flow Structures of Gaseous Jet Injected into Liquid for Underwater Propulsion," AIAA 2010-6911, 46th AIAA/ASME/SAE/ASEE Joint Propulsion Conference & Exhibit, July 2010, https://doi.org/10.2514/6.2010-6911.
To the best of my knowledge the authors should couple the numerical solution of the heat transfer equation in the bulk with the numerical solution of the heat conduction in the metallic surface (see my remark no 8).
6) What are the values of the k-e constants, which have been applied in the RANS turbulence model? For example what are the values of C1e and C2e?
7) (Line no. 143) How did the authors calculate the evaporation and condensation rates of the water? What is boiling regime applied in the evaporation process? Is it nucleate boiling?
8) (Eq. (7) –Line no. 169) it should be noted Johnson cook model (Eq. 7) is based on the surface temperature of the projectile. This model explicitly takes into account the hardening due to strain and strain rate, and also thermal softening due to high temperature caused by adiabatic heating. What is the value of the surface temperature of the projectile? Does it have other effects on the thermal deformation (such as: coefficient of thermal expansion, thermal fatigue) of metallic structure of the projectile?
9) Grid sensitivity calculation should be performed (see also figure 1).
10) (Section 4 Results and Discussion) the convergence plot of the CFD simulation should be added to this section.
11) The conclusion section needs to be extended. It should include the main results of this work and novelty of this research.
12) (References section) the digital object identifier (DOI) of the paper should appear:
Author 1, A.B.; Author 2, C.D. Title of the article. Abbreviated Journal Name Year, Volume, page range, DOI.
Author Response
Dear Reviewer,
Thank you for the comments, they are helpful for me to perfect this manuscript (Manuscript ID: energies-1592501). I have revised this manuscript according your comments, the modifications are marked by using the “Track Changes” function of MS Word. In the attachment, the corresponding explanation and rebuttal are below each question, the questions are marked with blue, the answers are written with black.
Hope my explanations can answer your questions. Please do not hesitate to contact me if you need further clarification.
Chuang Huang
March 1, 2022

Round 2
Reviewer 2 Report
Dear authors,
It is recommended to consider this interesting manuscript for publication.